# Human Glucose Transporters in Renal Glucose Homeostasis

**DOI:** 10.3390/ijms222413522

**Published:** 2021-12-16

**Authors:** Aleksandra Sędzikowska, Leszek Szablewski

**Affiliations:** Chair and Department of General Biology and Parasitology, Medical University of Warsaw, Chalubinskiego 5, 02-004 Warsaw, Poland; aleksandra.sedzikowska@wum.edu.pl

**Keywords:** kidney, glucose transporters, glucose homeostasis, physiology, diseases

## Abstract

The kidney plays an important role in glucose homeostasis by releasing glucose into the blood stream to prevent hypoglycemia. It is also responsible for the filtration and subsequent reabsorption or excretion of glucose. As glucose is hydrophilic and soluble in water, it is unable to pass through the lipid bilayer on its own; therefore, transport takes place using carrier proteins localized to the plasma membrane. Both sodium-independent glucose transporters (GLUT proteins) and sodium-dependent glucose transporters (SGLT proteins) are expressed in kidney tissue, and mutations of the genes coding for these glucose transporters lead to renal disorders and diseases, including renal cancers. In addition, several diseases may disturb the expression and/or function of renal glucose transporters. The aim of this review is to describe the role of the kidney in glucose homeostasis and the contribution of glucose transporters in renal physiology and renal diseases.

## 1. Introduction

The kidneys filter large quantities of plasma, and, in doing so, reabsorb substances that are necessary for the human body and secrete others destined for elimination. These renal functions play an important role in the regulation of fluid, fluid osmolarity, excretion of metabolic waste and foreign chemicals, arterial pressure, and hormone secretion. The kidneys also influence pH homeostasis and electrolyte balance [1], and are responsible for glucose balance.

Glucose homeostasis is governed by several processes, including glucose absorption in the gastrointestinal tract, its uptake by muscle and adipocytes, gluconeogenesis in the liver and kidneys, and its reabsorption and excretion by the liver and kidneys [1]. It is essential to maintain glucose homeostasis to prevent the pathological consequences resulting from long-lasting hyperglycemia, such as vascular complications, nephropathy, neuropathy, and retinopathy, as well as prolonged hypoglycemia, which may damage the central nervous system with potentially fatal results [2].

The lipid bilayer forming the plasma membrane is hydrophilic and, hence, impermeable for glucose; as such, glucose transport into cells depends on carrier proteins localized to the plasma membrane. Three families of glucose transporters have been identified in humans: GLUT proteins, encoded by *SLC2A* genes, sodium-dependent glucose transporters (SGLT proteins), encoded by *SLC5A* genes, and SWEET, encoded by the *SLC50A* gene [3]. Changes in the expression and/or function of these glucose transporters may lead to various renal disorders. In addition, renal diseases, type 2 diabetes mellitus, or cancers may disturb the expression and function of renal glucose transporters, and these may also be a target in the therapy of several diseases.

## 2. Glucose Metabolism in the Healthy Kidney

The mechanisms used to maintain glucose homeostasis are crucial in preventing the pathological consequences of hyperglycemia or hypoglycemia. These include the release of glucose into the circulation, uptake of glucose from the circulation, and reabsorption of glucose from the urine, which are performed by the kidneys. They also play key roles in the synthesis and secretion of hormones (e.g., renin, prostaglandins, kinins, erythropoietin), hydroxylation of vitamin D_3_ and gluconeogenesis, as well as the metabolism of several endogenous compounds, such as insulin and steroids [1].

### Overview of Renal Physiology

The kidneys filter large amounts of glucose. Reabsorption takes place in the proximal tubule, which retains most of the filtered glucose. Human kidneys produce a total of approximately 120 mL/min of ultrafiltrate, of which only 0.3–16 mL/min of urine is produced. In healthy humans, who have a normal estimated glomerular filtration rate (eGFR) of 180 l over 24 h, the kidneys filter about 180 g of glucose. The filtered glucose is reabsorbed in the proximal tubule, from which it is taken up into the peritubular capillaries and either returned to the systemic circulation or moved to the distal tubular segment as a source of energy [4]. Therefore, the urine of a healthy human is nearly free of glucose [5]. In healthy adult humans, the maximum renal glucose capacity, known as the tubular maximum for glucose (TmG), is about 375 mg/min [6]. The rate at which glucose is filtered is typically significantly lower than the TmG: i.e., for 180 g of glucose/day, this rate is 125 mg/min. Once the TmG is reached and the glucose transporters are unable to reabsorb all the glucose, glucosuria is observed.

The liver produces 80% of endogenous derived glucose, with the remaining 20% coming from the kidneys, which also contain gluconeogenic enzymes [7]. The process of renal gluconeogenesis occurs in the proximal tubule.

Renal gluconeogenesis synthesizes glucose at the rate at which it is consumed by the kidney. During fasting, for example, during the night, the kidney produces ~2 mg/kg/min of glucose, which corresponds to the amount of glucose absorbed by the tissues [7]. The primary substrates for renal gluconeogenesis are lactate, glutamine, glycerol, and alanine [8]. However, the primary precursor for renal gluconeogenesis is lactate, and the main gluconeogenic amino acid in the kidney is glutamine; renal gluconeogenesis from lactate is 3.5-, 2.5-, and 9.6-fold faster compared to glycerol, glutamine, and alanine, respectively [8]. Renal gluconeogenesis is performed through the glucose–glutamine cycle [9]. Glutamine-based gluconeogenesis produces ammonia, resulting in enhanced excretion of hydrogen ions in the form of ammonium (NH_4_^+^); this may help control metabolic acidosis in the kidney [10].

## 3. Glucose Transporters in Healthy Human Kidneys

As mentioned earlier, the transport of glucose across the plasma membrane depends on membrane proteins, called glucose transporters. These proteins belong to the major facilitator superfamily (MFS), which contains 74 families of membrane transporters.

While more than 10,000 membrane transporters have been sequenced, the largest individual group within the MFS is the sugar porter family. Sugar porters are widely distributed between bacteria, archaea, and eukaryotes [11]. Glucose transporters are ubiquitously expressed and are highly conserved from bacteria to humans. However, although these proteins are called glucose transporters, they may transport different substrates, such as ions (I^−^, ClO_4_^−^, SCN^−^, NO_3_^−^, Br^−^), lactate, pyruvate, *myo*-inositol, urate, vitamins (dehydroascorbic acid, nicotinate, pantothenic acid, biotic, α-lipoic acid), and monosaccharides other than glucose (galactose, fructose, fucose, xylose), and glucosamine. In addition, these transporters may also act as glucose sensors, and the gene that codes them may also act as an autoimmune modifier.

In humans, glucose transporters are encoded by three distinct families of genes: *SLC2A, SLC5A,* and *SLC50A* [3], which code for three different classes of glucose transporters: sodium-independent glucose transporters (facilitated transport, GLUT proteins, and *SLC2A* genes), sodium-dependent glucose symporters (secondary active transport, SGLT proteins, and *SLC5A* genes), and a new class of glucose uniporters (SWEET proteins, *SLC50A* genes). Most cells express more than one glucose transporter, and their expression depends on the specific metabolic requirements.

### 3.1. Characteristics of Human Glucose Transporters

In the healthy human kidney, two classes of glucose transporters are expressed and are involved in renal glucose transport: GLUT proteins and SGLT proteins. To date, no SWEET proteins have been detected in human kidney tissue.

Fourteen GLUTs, GLUT1–GLUT14, have been identified in humans [12], and these are encoded by the solute-linked carrier family 2, subfamily A genes *SLC2A1–SLC2A14* [13]. All GLUT proteins contain 12 hydrophilic membranes spanning α-helical transmembrane (TM) domains. These are connected by the hydrophilic loop between TM6 and TM7 of the protein [14]. GLUT proteins also contain a short intracellular N-terminal segment, a large C-terminal segment, and a single site for glycosylation on the exofacial end. The glycosylation site may be localized to the large loop between TM1 and TM2 or between TM9 and TM10 [15]. Based on the phylogenetic analysis of sequence similarity, the GLUT proteins are divided into three classes [16]: Class I, including GLUT1–GLUT4 and GLUT14, Class II, including GLUT5, GLUT7, GLUT9, and GLUT11, and Class III, including GLUT6, GLUT8, GLUT10, GLUT12, and GLUT13 (HMIT). All human GLUT proteins are facilitative transporters, except for GLUT13 (HMIT), which is an H^+^/*myo*-inositol symporter [17].

The sodium-dependent glucose co-transporters are the members of a large gene family (*SLC5A*), the SGLTs or sodium/substrate symporters family (SSSF), which contains over 450 members [18]. Considerable diversity has been observed in the genes of these symporters, for example, in the number of exons. The symporters themselves contain 14 TM α-helices, apart from the sodium-iodide symporter (NIS) and SMCT, which lack TM^14^ [19]. The hydrophilic N- and C-termini are located on the extracellular side of the cell membrane [3]. Although SGLTs are highly-glycosylated membrane proteins, this glycosylation is not needed for their operation. Twelve *SLC5A* family genes have been identified in humans. All of them code for sodium/glucose co-transporters, except SGLT3 (*SLC5A4*), which acts as a glucose sensor [20].

#### 3.1.1. Human Renal GLUT Proteins

Ten GLUT proteins have been identified in healthy human kidneys: GLUT1, GLUT2, GLUT4, GLUT5, and GLUT8–GLUT13 (Table 1).

GLUT1 is localized to the basolateral membrane of the epithelial cells in the S3 segment of the proximal tubules [6], where it is associated with SGLT1; however, it can be found at various levels in all segments of the nephron [8]. GLUT1 releases glucose reabsorbed by SGLT1 back into the bloodstream [21]. As GLUT1 is localized along the entire length of the proximal tubule, it has been proposed that this transporter is involved in transcellular glucose transport in the S3 segments [22]. In contrast, in rat kidneys, the strongest GLUT1 expression is detected in the basolateral distal tubule segments, including the medullary thin and thick ascending limbs; the highest levels are observed in the connecting segments and connecting ducts, especially in principal cells and in intracalated cells [23]. A correlation has been noted between the expression of GLUT1 and the glycolytic activity of the nephrons and collecting duct segments. Therefore, it has been suggested that the most distal tubule segments take up glucose as a source of energy via basolateral GLUT1 [5].

GLUT2 is located at the basolateral membrane of the epithelial renal tubule [22]. It is strongly expressed in the convoluted tubule (S1/S2 segments) and at the lower levels of the proximal straight tubule (S3 segment) [24,25]. GLUT2 facilitates glucose efflux back into the blood, which is reabsorbed by SGLT2 [21,26]. It also releases fructose into the blood, which is reabsorbed by GLUT5. Various precursors are generated in the proximal tubule by glucose-6-phosphatase, which forms free glucose. The generated glucose may then exit the cell via GLUT2 across the basolateral membrane. In this way, healthy human kidneys generate approximately 15 to 55 g of glucose per day, particularly during fasting [5].

Other GLUT proteins have been detected in the human kidney; however, little is known of their functions in this location.

GLUT4 is expressed in the glomerulus [27], and in the mesangial cells and podocytes. Animal studies indicate that GLUT1, GLUT2, and GLUT4 expression in podocytes is modulated by angiotensin II [28]. The presence of the GLUT4 mRNA and protein were detected in the thick ascending limb of the loop of Henle (TAL), and it has been suggested that GLUT4 may be involved in managing the local resources in this segment [24].

GLUT5 has been found to be expressed in the apical membrane of proximal straight tubule (S3 segment) in animal studies. It primarily acts as a fructose transporter [29]. The physiological concentration of fructose is estimated to range from 0.008 to 0.03 mM in the blood, and, presumably, the glomerular filtrate, and 0.035 mM in the urine of healthy humans [30]. Differences in the concentration of fructose determine the direction of fructose flux managed by GLUT5.

GLUT8 is expressed in the glomerulus and in the podocytes [27]; however, literature on its role in the kidneys is scarce [31].

GLUT9 is expressed in the proximal tubule in the epithelial cells; GLUT9a is detected in the basolateral membrane, and GLUT9b on the apical pole. These isoforms are also uric acid transporters [21].

GLUT10 mRNA was detected at low levels in the kidneys, but its role in the kidneys needs further investigation [32].

GLUT11 is known in three isoforms: GLUT11-A, GLUT11-B, and GLUT11-C. Of these, isoforms GLUT11-A and GLUT11-B have been detected in human kidneys; however, their role there remains unclear [33].

GLUT12 appears to be expressed in the apical membrane of tubules and collecting ducts, as indicated by animal studies [34]. Unfortunately, its functional contribution in human kidneys is unknown.

GLUT13 (HMIT), a H^+^/*myo*-inositol transporter, was also detected in kidney tissue. However, while its role in the brain is well described, its role in the kidneys needs further investigation [33,35].

**Table 1 ijms-22-13522-t001:** Characteristics of sodium-independent glucose transporters (GLUT proteins) in the healthy human kidneys.

Glucose Transporter	Gene	Characteristics	References
GLUT1	*SLC2A1*	Expressed at the basolateral membrane of the epithelial cells in the S3 segment of the proximal tubule. GLUT1 releases glucose reabsorbed earlier by SGLT1 into the bloodstream.	[7,8,13,21,27]
GLUT2	*SLC2A2*	Expressed at the basolateral membrane of the epithelial renal tubules. GLUT2 releases glucose, reabsorbed earlier by SGLT2, and fructose, reabsorbed by GLUT5, into the bloodstream.	[13,21,22,23,26]
GLUT4	*SLC2A4*	Expressed in the glomerulus, mesangial cells, and podocytes. Its role in the human kidneys needs further investigation.	[13,27]
GLUT5	*SLC2A5*	Expressed in the apical plasma membrane in the S3 segment of the proximal tubule cells. GLUT5 reabsorbs fructose from urine.	[13,29]
GLUT8	*SLC2A8*	Expressed in the glomerulus and in podocytes. Its precise role in human kidneys needs further investigation.	[13,27]
GLUT9 (earlier designated as GLUTX)	*SLC2A9*	GLUT9 is expressed in the epithelial cells of proximal tubule; GLUT9a in the basolateral membrane; GLUT9b is localized on the apical pole. It is involved in transport of uric acid.	[13,15,21]
GLUT10	*SLC2A10*	GLUT10 mRNA is detected at low levels in kidney. Its role in kidney remains unclear.	[13,32]
GLUT11 (earlier designated as GLUT10)	*SLC2A11*	In the human kidneys are expressed GLUT11-A and GLUT11-B isoforms, lack of expression of GLUT11-C isoform. Their role in human kidneys needs further investigation.	[13,33]
GLUT12 (earlier designated as GLUT8)	*SLC2A12*	Its expression was described in the renal distal tubules and collecting ducts in animal models of hypertension and diabetic nephropathy.	[13,34]
GLUT13 (HMIT)	*SLC2A13*	An H^+^/*myo*-inositol co-transporter that exhibits transport activity only for *myo*-inositol. Its function in kidney remains unknown.	[13,34,35]

#### 3.1.2. Human Renal Sodium-Dependent Glucose Co-Transporters

Nine sodium-dependent co-transporters are expressed in the human kidney: SGLT1–SGLT6, SMIT1, SMVT, and SMCT1 (Table 2).

SGLT1 is expressed on the luminal surface of cells within the S2/S3 segment of the most distal position of the proximal tubule [36]. In human kidneys, stronger SGLT1 expression was detected in the S3 segment than in the S2 segment [37]. SGLT1 expression was also found in the luminal membrane of the TAL and in the macula densa (MD) [37,38,39]. The protein is believed to contribute to renal glucose reabsorption from the urine. It also has a minor role in the renal reabsorption of glucose [40], contributing to 10 to 20% of the glucose reabsorbed in the proximal tubule [1]. The absorbed glucose is then released into the circulation via GLUT1 [41]. SGLT1 transports Na^+^ and glucose at a coupling ration of 2:1 [42]. Besides the sodium ions and glucose, SGLT1 can also transport 264 molecules of water [43].

SGLT2 is highly expressed in the kidney cortex and has been identified as a kidney-specific transporter. It is localized on the apical domain of the epithelial cells of the proximal tubule (S1/S2 segments) [3,21]. SGLT2 is responsible for 90% of glucose filtered at the glomeruli [40], i.e., about 160 g/24 h [44,45]. Together with SGLT1, it controls the initial step of renal glucose reabsorption. The stoichiometry of SGLT2 is 1 Na^+^: 1 sugar [46].

SGLT3 is expressed in human proximal tubular cells. Studies based on the human proximal HK-2 kidney cell line suggest that it may be responsible for sodium reabsorption. The sodium-to-sugar stoichiometry is 2:1 [47].

SGLT4 is expressed in human kidney tissue, but few studies have been carried out regarding its function [3,18]. It is known to transport mannose, glucose, fructose, and galactose, and it has been proposed that SGLT4 is responsible for renal reabsorption of mannose and may be involved in mannose homeostasis [48].

SGLT5 is expressed in the human kidney cortex; however, its precise localization and function remain unknown. It is believed to be a kidney-specific sodium dependent mannose transporter and may act as a complementary mannose transporter that regulates renal reabsorption of mannose. Studies suggest that it also transports glucose and fructose [49].

SGLT6, now known as SMIT2, is detected in the luminal side of the proximal convoluted tubules in the kidneys of rabbits [50]. The human *SLC5A11* gene, which codes SMIT2, interacts with various immune-related genes and may play a role as an autoimmune modifier [51].

Literature regarding the three remaining sodium-dependent glucose co-transporters expressed in human kidneys, SMIT1, SMVT, and SMCT1, is scarce, particularly concerning their subcellular localization and physiological role.

SMIT1 is an Na^+^/*myo*-inositol co-transporter [52]. It is present in three transcript variants formed by splicing within and distal to exon two: SMIT1a, SMIT1b, and SMIT1c. Of these, SMIT1b and SMIT1c lack TM^14^ [3,21].

SMVT is a multivitamin co-transporter [3] that transports pantothenic acid, biotin, and α-lipoic acid. It is also a sodium/iodide co-transporter [53].

SMCT1, also known as AIT, transports short chain fatty acids, lactate, pyruvate, and nicotinate, with a stoichiometry of 2:1 [54,55].

**Table 2 ijms-22-13522-t002:** Characteristics of sodium-dependent glucose transporters (SGLT proteins) in the healthy human kidneys.

Glucose Transporter	Gene	Characteristics	References
SGLT1	*SLC5A1*	Expressed in the S3 segment of the proximal tubule on the luminal surface of the kidney cells. It reabsorbs glucose from urine, which is then released into the circulation by GLUT1.	[1,13,21,22,40,41,45]
SGLT2	*SLC5A2*	Expressed in the S1 and S2 segments of the proximal tubule on the luminal surface of the epithelial cells. SGLT2 plays a major role in the reabsorption of glucose from urine, which is then released into the circulation via GLUT1.	[13,22,46]
SGLT3	*SLC5A4*	Expressed in the proximal tubule of the human kidneys. Little is known about its expression and function. It may be responsible for the reabsorption of sodium.	[13,47]
SGLT4	*SLC5A9*	Expressed in human kidney tissue, but few studies have been carried out regarding its function. It may be responsible for the reabsorption of mannose, and may be involved in mannose homeostasis.	[3,18,48]
SGLT5	*SLC5A10*	Expressed in the human kidney cortex. Its physiological role in the kidney remains unknown.	[13,49]
SGLT6 (SMIT2)	*SLC5A11*	Its precise localization and function in human kidney remain unknown and need further investigation.	[13,50]
SMIT1	*SLC5A3*	An Na^+^/*myo*-inositol co-transporter; however, its precise localization and function need further investigation.	[13,21,52]
SMVT	*SLC5A6*	Multivitamin co-transporter. Its precise localization and function in the kidney need further investigation.	[3,13,53]
SMCT1 (AIT)	*SLC5A8*	Expressed in the kidney, but its precise localization and function need further investigation.	[13,54,55]

## 4. Renal Diseases Associated with Changes in Expression of Glucose Transporters

Several pathological renal conditions are known to exist, which may be caused by various factors, such as bacterial infection, parasites, inherited anatomical pathologies, and structural injury. Renal disorders may also cause changes in the expression of glucose transporters, or may themselves be the effects of these changes (Table 2).

### 4.1. Glucose–Galactose Malabsorption

Glucose–galactose malabsorption (GGM) occurs due to mutations in the *SLC5A1* gene coding for the SGLT1 sodium-dependent glucose co-transporter. It is a rare inherited autosomal recessive disease [56]. Patients with GGM have been found to carry various missense, nonsense, frame-shift, and specific-site mutations in the SGLT gene. These mutations impair the insertion of SGLT1 into the enterocyte and tubular membrane [57], and cases have been noted where non-functional SGLT1 is present within the apical plasma membrane [58]. Missense mutations, i.e., those that change a single amino acid, result in incorrect sorting of the protein in cells; as such, it is suggested that conformational changes in the protein may interfere with the proper folding and/or delivery of the sodium-dependent glucose co-transporter into the plasma membrane, as well as its integrity, thus affecting its function [21].

GGM was first described in 1962 [59] as severe life-threating watery diarrhea in neonates [45]. It is fatal within weeks unless lactose, glucose, and galactose are removed from the diet; the diarrhea returns immediately upon the reintroduction of these sugars. Patients with GGM may demonstrate mild and absent renal glucosuria [43,60], as well as nephrolithiasis and nephrocalcinosis caused by chronic dehydration. In addition, nephrocalcinosis may also be caused by the dysfunction of the renal tubule, resulting in hypercalcemia, metabolic acidosis, and dehydration [45,61].

### 4.2. Familial Renal Glucosuria

Familial renal glucosuria (FRG) is a rare renal tubular disorder caused by autosomal recessive mutations in gene *SLC5A2* coding the SGLT2 sodium-dependent glucose co-transporter. The first *SLC5A2* gene mutation was described in 2002 [62]. To date, about fifty mutations have been identified, most of these being private, such as premature stops, frame shifts, missense mutations, and splicing mutations. Private mutation is a rare gene mutation that is usually found only in a single family or a small population. A private mutation occurs and is passed to a few family members but not to future generations. Studies indicate that intron 7 is a mutational hot spot [60]. Patients with similar or even identical mutations show a broad range of severity in glucosuria, suggesting that environmental factors also affect urinary glucose reabsorption [21]. Decreased glucose reabsorption results in the excretion of glucose through the urine without affecting other glomerular tubular renal functions [63]. As most patients with FRG do not demonstrate any clinical manifestations, it is commonly described as a “nondisease” and is treated as benign glucosuria.

The excretion of glucose in patients with FRG ranges from 1 to 162 g/1.73 m^2^/24 h [64,65] and is dependent on the genotype. In humans, mild glucosuria (<10 g/1.73 m^2^/24 h) tends to be associated with heterozygous *SLC5A2*, while severe glucosuria (≥10 g/1.73 m^2^/24 h) is experienced by individuals with homozygosity or compound heterozygosity for SGLT2 mutations [65]. Interestingly, FRG has also been diagnosed in a patient without any mutations in the *SLC5A2* gene [66].

Animal studies suggest that a lack of SGLT2 causes increased urine output and significantly increased glucosuria. This is associated with compensatory increased feeding, drinking, and activity [21].

### 4.3. Fanconi-Bickel Syndrome

Fanconi–Bickel Syndrome (FBS) is an extremely rare syndrome. The first patient with FBS was described in 1949 [67], and, until 2002, only 112 patients have been reported [68]. Homozygous or compound heterozygous mutations in the *SLC2A2* gene, which codes GLUT2, are believed to be responsible for a glycogen storage disease (GSD) termed GSD XI [21,69]. This syndrome, originally named hepatorenal glycogenosis with renal Fanconi syndrome, has been attributed to autosomal, recessive mutations in the *SLC2A2* gene. While most patients with FBS are homozygous, some are compound heterozygous [70]. Younger neonates tend to demonstrate failure to thrive, polydipsia, and constipation, while older children are more likely to demonstrate osteopenia, short stature, hepatomegaly, tubular nephropathy with associated glycosuria, phosphaturia, aminoaciduria, and intermittent proteinuria [71]. Impaired glucose transport from the tubular cells results in greater glycogen and glucose accumulation within these cells [72]. Patients with FBS also demonstrate hepatomegaly secondary to glycogen accumulation [73].

### 4.4. Urate Metabolism Disorders

The urate metabolism is regulated by GLUT9. Urate is secreted into the blood by the liver, from which it is later reabsorbed by the kidneys into the proximal convoluted tubule [12]. Urate is transferred across the renal epithelium by URAT1, which is expressed in the apical membrane, and GLUT9a, expressed in the basolateral membrane. Significant relationships have been identified between the expression of GLUT9, serum uric acid levels, and gout [74], as well as a positive correlation between the levels of GLUT9 mRNA and serum uric acid concentration [75].

GLUT9 loss in function caused by mutations in the *SLC2A9* gene results in hypouricemia, possibly caused by the reduced release of urate from the liver and impaired reabsorption of urate from the urine [12,76]. Homozygous mutations in the gene coding for GLUT9 can completely block uric acid absorption, resulting in severe hypouricemia complicated by nephrolithiasis and exercise-induced acute renal failure [77]. Hypouricemia may also be caused by heterozygous missense mutations in GLUT9a [78] or by the increased expression of GLUT 9 [12,21]. Animal studies indicate that the loss of GLUT9 function in mice causes hyperuricemia, hyperuricosuria, and early onset nephropathy [21,74]; in contrast, the dysfunction of GLUT9 in humans is associated with hypouricemia. It is suggested that hyperuricemia in mice caused by the loss of GLUT9 function may occur as a result of the impaired uptake of uric acid by the liver and an inability to degrade it to allantoin by uricase [21].

### 4.5. Severe Inflammation

Acute renal failure (ARF) is defined as an abrupt decline in the glomerular filtrate rate (GFR) [79]. The important risk factors of ARF are sepsis and septic shock, and the reduction of GFR associated with sepsis is secondary to altered hemodynamics [80]. Few studies, however, have examined the pathophysiology of renal tubular dysfunction with the failure in urine concentration and increased fractional excretion of glucose with glucosuria due to sepsis.

Animal studies indicate that endotoxemia changes the expression of glucose transporters in the kidney, resulting in impaired GFR. The results suggest that cytokines affect the expression of tubular glucose transporters [81]. Elsewhere, in a mouse model, it has been found that the cytokines produced during sepsis, such as tumor necrosis factor-α (TNF-α), interleukin-β (IL-β), and interferon-γ (IFN-γ), significantly decrease the expression of SGLT2, SGLT3, and GLUT2 and significantly increase that of SGLT1 and GLUT1 [79]. The findings suggest that GFR is impaired by the altered glucose transporter expression caused by the cytokines produced during sepsis.

### 4.6. Diabetic Kidney Disease

Diabetic kidney disease (DKD) is the serious complication of diabetes and common cause of end-stage renal disease (ESRD) worldwide. ESRD may occur in response to either type 1 diabetes mellitus (T1DM) or type 2 diabetes mellitus (T2DM). Unfortunately, treatments targeting hyperglycemia, blood pressure, and lifestyle do not appear to slow the progression of this disease. In addition, the mechanisms causing the development of DKD remain poorly understood [80].

Nephropathy is one of the major microvascular complications of diabetes caused by long-lasting hyperglycemia. Patients with diabetes may demonstrate chronic kidney disease (CKD), known as chronic renal failure (CRF) [81], i.e., the permanent loss of nephron units; this can prompt the induction of compensatory mechanisms, glomerular hyperfiltration, and tubular hypertrophy. These pathologies, in turn, may cause glomerular sclerosis with progression to ESRD [82].

Patients with diabetes demonstrate the three-fold greater reabsorption of glucose in comparison with healthy subjects [83]. Long-lasting hyperglycemia in patients with diabetes increases the expression of GLUTs in the proximal tubule, and, thus, increased glucose reabsorption in the proximal tubule. The type of diabetes influences the prevalence of glomerular hyperfiltration; studies indicate this value to be about 13 to 75% in patients with T1DM and 0 to 40% in those with T2DM [84].

Although diabetes is known to alter the expression and activity of renal sodium-dependent glucose co-transporters, little information exists on the expression of SGLT1 in the diabetic kidney. Increased levels of renal SGLT1 were observed in a mouse model of T2DM [85]; however, in a mouse model of T1DM, renal SGLT1 has been found to be either upregulated or downregulated based on the study [86]. Elsewhere, no significant difference in the SGLT1 level or nephropathy were observed between fresh biopsies of kidneys from patients with T2DM and non-diabetic controls [87]. In HEK-293 T cells, insulin decreased glucose transport mediated by SGLT1 [88]. In a diabetic Zucker rat model, long-lasting hyperglycemia resulted in no changes in SGLT1 expression in one study [89], while elevated SGLT1 mRNA during diabetes was observed in another [90]. In addition, the markedly increased expression of SGLT1 mRNA was observed in kidney biopsy specimens obtained from patients with diabetes, and the SGLT1 mRNA level significantly correlated with fasting and postprandial plasma glucose and HbA1c [91]. In addition, various studies have identified different levels of expression of SGLT1 in diabetic kidney tissue [4].

More studies have been performed on the expression of SGLT2 in diabetic kidneys; however, they have yielded ambiguous findings. The data on glucose co-transporter expression in patients with diabetes is sparse and varied [4]. Studies on human exofoliated proximal tubular epithelial cells (HEPTEC) isolated from fresh urine found significantly higher expression of SGLT2 in patients with T2DM compared to healthy controls [92]. These observations confirmed previous findings in animal studies [93]. In addition, elevated levels of SGLT2 mRNA have been reported in animal models of diabetes [90]. It has, therefore, been concluded that diabetic renal tubular and glomerular disease may be associated with SGLT2 overexpression. While SGLT2 mRNA levels have been found to be downregulated in patients with type 2 diabetes, this change is not statistically significant [91]. These findings contradict those of a previous study in which the level of SGLT2 mRNA was found to be approximately four-fold higher in cultured tubular cells isolated from the urine of patients with T2DM compared to healthy controls [92].

In addition, a decrease in SGLT2 mRNA and protein has been observed in patients with T2DM [94], whereas another study found increased SGLT2 to be associated with nephropathy [87]. It has been suggested that the differences observed between these human studies may be due to different methods in obtaining human kidney tissue and the mode of selected patients with T2DM and control subjects [91]. While several other results have been obtained regarding the expression of SGLT2 in patients with diabetes, these are often variable and sometimes contradictory [4].

The co-transporter SGLT3 is expressed in human proximal tubular cells [47]. It does not transport glucose but facilitates the influx of Na^+^ in the presence of extracellular glucose [95]. In patients with diabetes, enhanced sodium reabsorption has been observed in the proximal tubule [96]. Sodium resorption in the proximal tubule has been found to increase by about 20% in children with T1DM and adults with T2DM compared to healthy controls [97,98]. Studies have also reported SGLT3 upregulation in COS-7 cells and HK-2 cells (mammalian kidney-derived cells). SGLT 3 overexpression increases the intracellular concentration of Na^+^ in COS-7 cells by three-fold and in HK-2 cells by 5.5-fold; the researchers concluded that SGLT3 upregulation enhances Na^+^ absorption in the proximal tubule, resulting in hyperfiltration and renal injury [47].

Changes in the expression of sodium-independent glucose transporter (GLUT) proteins were also observed in diabetic kidney tissue. However, while the expression of GLUT1 mRNA is downregulated in the glomeruli of normoalbuminuric patients with T1DM, it is upregulated in patients with microalbuminuria as compared to healthy controls. These results have been confirmed in animal studies [36]. Hence, it is suggested that, while the overexpression of GLUT1 in mesangial cells is deleterious, its expression in podocytes protects against diabetic kidney disease [36]. Other animal studies have demonstrated decreased levels of the GLUT1 protein and GLUT1 mRNA [5,21]. Other studies suggest that GLUT1 overexpression in mesangial cells may cause the development of nephropathy with diabetes [99,100]. In addition, genetic variations of *SLC2A1* also affect nephropathy and may be involved in the risk of micro- and macroalbuminuria in adult European Americans patients with T2DM [101].

The increased expression of GLUT2 and its translocation to the luminal surface of the proximal tubules has been noted in an animal model of diabetes [102]. GLUT2 overexpression was also observed in HEPTECs isolated from the urine of patients with T2DM, and GLUT2 expression at the proximal tubule of the brush border membrane depends on the level of blood glucose, which may be changed by diabetic nephropathy. The level of GLUT2 has been found to be 30-fold higher as compared to healthy controls [92]. Interestingly, hyperglycemia has been found to induce damage to mesangial cells.

In addition, increased levels of the GLUT5 protein and mRNA have been identified in STZ-induced diabetic rats at the proximal tubule brush border membrane [102].

An association has also been noted between decreased GLUT4 levels in skeletal muscle with insulin resistance in chronic kidney disease, which may stimulate the development of T2DM [103].

### 4.7. Renal Cancers

Several different histologically and genetically distinct types of renal cell carcinoma (RCC) have been found. According to the Heidelberg classification, the main subtypes are clear cell RCC (ccRCC), papillary RCC, chromophobe RCC, oncocytoma, and collecting duct carcinoma [104]; however, other subtypes have also been proposed [105]. The most common type of RCC is clear cell carcinoma. It is characterized by clear cytoplasmic cells containing glycogen.

Many cancers demonstrate altered glucose transporter expression, with the expression depending on both the type of RCC and kind of glucose transporter. It has also been noted that GLUT upregulation is associated with reduced patient survival.

A highly-significant correlation has been observed between the presence of a polymorphism in the *SLC2A1* gene, the gene that codes GLUT1, and the development of ccRCC [106]. GLUT1 levels are higher in renal samples obtained from ccRCC patients compared to those from healthy subjects; this is also true for chromophobe RCC and papillary RCC [107]. In addition, the cytoplasmic expression of GLUT1 was found to correlate with the membranous expression of GLUT1 in all the investigated subtypes of renal cell carcinoma. In this research, the highest expression in the cell membrane was observed in the case of GLUT1. High membranous expression of GLUT1 was detected in 44.7% of ccRCC patients, moderate expression in 26.5%, and no expression in 27% of the patients [108]. In addition, negative GLUT1 expression was observed in 75.8% of the patients with papillary RCC, and 71.5% with chromophobe RCC [108].

Strong immunohistochemical staining for GLUT1 was observed in ccRCC tissues, predominantly in the cell membrane of the tumor cells but also partially in the cytoplasm [109]. Interestingly, in normal kidney tissue, GLUT1 is expressed primarily in the cytoplasm. Elsewhere, GLUT1 was found to be negative in 75.8% of the samples obtained from patients with papillary RCC, and in 71.5% of the samples obtained from patients with chromophobe RCC [108]. In another study, membranous expression was noted in 86.2% of patients with ccRCC and in 100% of patients with transitional cell carcinoma; however, no membranous expression was observed in other subtypes [110]. In addition, cytoplasmic expression of GLUT1 was noted in 55.2% of patients with ccRCC, 38% with papillary RCC, 13% with chromophobe RCC, 22% with oncocytomas, and in 82% of the patients with transitional cell carcinoma [110].

Regarding the relationship between GLUT1 expression and RCC type, correlations have been reported between the level of GLUT1 expression and the VHL tumor suppressor gene [105]. GLUT1 was overexpressed in cells lacking VHL due to mutations and was underexpressed in cells with VHL [111]. It has been suggested that no significant correlation exists between GLUT1 expression and tumor grade or tumor stage, nor between GLUT1 expression and patient survival [110]; however, other studies suggest a significant correlation between expression and OS [112]. Although GLUT1 may be a target for anticancer therapy in most ccRCC, it does not seem to be a prognostic marker for RCC [110].

GLUT2 was found to be downregulated in ccRCC [109,113]: one study found its level to be decreased by around nine-fold in 10 out of the 11 tumor samples tested [114]. Interestingly, GLUT2 downregulation was noted in chromophobe RCC but not in oncocytoma [113]. The significant downregulation of GLUT2 mRNA was observed in the case of patients with ccRCC, oncocytoma, and renal B-lymphoma [114].

Little information exists on the expression of other GLUT proteins in renal cancers. Increased levels of GLUT3 mRNA (fold of increase 8.0 ± 1.5) were observed in seven of the eleven patients with ccRCC, while no change was observed in the other four [114]. Elsewhere, GLUT3 expression was observed in 37.6% of the RCC samples, and immunostaining revealed weak staining in the cell membrane and granular staining in the cytoplasm [112].

However, the data regarding GLUT4 are ambiguous. One study noted a decreased level of GLUT4 in patients with ccRCC and upregulation in the case of patients with chromophobe RCC [113], while another reported GLUT4 overexpression only in patients with stage 4 RCC [112].

GLUT5 is overexpressed in ccRCC to a higher level than noted in the chromophobe and papillary types [112]. As GLUT5 is a fructose transporter, it has been suggested that fructose may be an additional source of energy for cancer cells [112]. Strong GLUT5 immunostaining was detected in the cell membrane and cytoplasm of 57.6% of the RCC samples in one study, but no immunostaining of the vascular endothelium, glomerulus, and interstice region were noted [112]. Increased GLUT5 levels were detected in patients with pelvic invasion and capsule breakage during diagnosis, suggesting that GLUT5 may be correlated with grade II differentiation and aggressiveness, and could be involved in the development of RCC [112]. In contrast, another study found GLUT5 expression to be decreased in ccRCC, and no changes were observed in oncocytoma RCC compared to healthy kidney tissue [113].

GLUT9 and GLUT12 are downregulated in ccRCC [113].

Little is known about the expression of sodium-dependent glucose co-transporters in renal carcinoma. Observations performed on Japanese patients with ccRCC suggest that SGLT1 and SGLT2 are predominantly located in the cell membrane and partially in the cytoplasm [109]. In most cases, SGLT1 staining was found to be weak, with normal renal tissue showing stronger staining. In addition, three out of sixty-eight cancer tissue samples lacked SGLT 2 staining, and the remaining cases only demonstrated weak staining. As with SGLT1, stronger SGLT2 staining was observed in normal parenchyma.

It has, therefore, been proposed that increased SGLT2 expression is significantly associated with shorter OS regardless of metastatic status; however, no significant correlations were observed between the expression of SGLT1 and SGLT2 and conventional pathological variables. The authors suggest that SGLT2 expression may be a prognostic marker in human RCC and may serve as a possible therapeutic target in RCC [109].

### 4.8. Metabolic Syndrome

Two major aspects of metabolic syndrome (MetS) are hyperglycemia and hyperuricemia. While the kidneys play a crucial role in the homeostasis of glucose and uric acid, little data exist on the association between MetS and glucose transporter expression. However, some results have been obtained from animal studies.

In Sprague Dawley rats, metabolic syndrome was induced by a high fructose diet (60%) for three months (FR-3) and five months (FR-5) [115]. At the end of the study, significantly increased GLUT2 and SGLT2 were observed in kidney tissue, and the extent correlated with the length of diet. SGLT1 was increased significantly in both FR-3 (413.5 ± 88.3% of control) and FR-5 (677.6 ± 26.5% of control), as well as for SGLT2 in FR-3 (643.1 ± 41.3% of control), and FR-5 (563.21 ± 21.7%) of control values. The FR-3 group demonstrated comparable GLUT1 expression to the control group, whereas FR-5 animals demonstrated significantly lower expression (55 ± 6.5% of control). Considerable differences in the metabolism and renal handling were also observed between glucose and fructose [116]. In the renal tubule, the filtration rate of fructose is higher than that of glucose. Filtered glucose is reabsorbed via SGLT1 and SGLT2, while fructose reuptake is mainly through GLUT2 and GLUT5 [117]. These results suggest that the chronic consumption of fructose upregulates GLUT2, GLUT9, SGLT1, and SGLT9, implying that consumption may simultaneously enhance the reabsorption of glucose. Studies have noted an association between MetS and increased glucose transport that precedes the full manifestation of diabetes, and that changes in renal epithelial transport develop earlier than diabetes mellitus [115].

Elsewhere, MetS was induced by monosodium glutamate in rats for nine days. The expression of GLUT1 in the kidneys of the experimental rats was found to be ~13 times lower at three months, and its content was restored at six months. At six months of age, the experimental group demonstrated twice the level of GLUT2 in comparison with the control. The authors concluded that critical upregulation of GLUT2 may be involved in the development of kidney disease [118].

## 5. Role of Glucose Transporters as Potential Targets in Kidney Disease

Glucose transporters, sodium-independent GLUTs, and sodium-dependent SGLTs play important roles in renal physiology, and any disturbances in their expression and/or function are associated with kidney diseases. As such, these membrane proteins may be therapeutic targets in kidney diseases.

### 5.1. Diabetic Kidney

The SGLT2 sodium-glucose co-transporter is involved in glomerular filtration, where it reabsorbs about 90% of the glucose filtered in glomeruli. Therefore, it has been suggested that the use of SGLT 2 inhibitors may regulate glucose reabsorption in the kidneys.

In the late 1980s, it was found that the administration of phlorizin, first isolated in 1863 from apple tree bark [119], induces glucosuria and normalizes fasting and fed plasma glucose levels [120,121]. It was suggested that phlorizin blocks both SGLT1 and SGLT2 activity [45]. Unfortunately, it was also found that phlorizin lacks any therapeutic potential as a treatment for patients with diabetes due to its poor oral absorption and the fact that SGLT1 is an intestinal sodium-dependent glucose co-transporter, and, as such, its blockage causes diarrhea [122]. Therefore, phlorizin derivatives acting as selective inhibitors of SGLT2 have been designed. However, although these inhibitors have been approved as glucose lowering agents for patients with T2DM, they also reduce the single-nephron glomerular filtration rate (SNGFR) in chronically diseased kidneys [123]. As mentioned earlier, diabetic nephropathy is the leading cause of end-stage renal failure. When given chronically, the inhibitors markedly slow the progress of chronic kidney disease in patients with T2DM [122]. Several inhibitors have become available, such as Canagliflozin, Dapagliflozin, and Empagliflozin [124].

During the inhibition of SGLT2, patients with normal renal function tend to demonstrate similar changes in the glomerular filtration rate as patients with mild to moderate chronic kidney disease. It was also found that SGLT2 inhibitors lower the risk of congestive heart failure, a major cardiovascular complication in diabetic kidney disease, and reduce cardiovascular mortality in patients with T2DM and CKD. The therapeutic interactions between insulin and SGLT2 inhibitors have also been investigated [125]. The mechanisms of action of SGLT2 inhibitors and dosing recommendations are presented in reviews [5,126]. However, the long-term effects of the administration of SGLT2 inhibitors remain unknown; for example, an increased occurrence of urogenital infections has been observed in women following therapy with SGLT2 inhibitors [127].

Incretin-based drugs have also been proposed for declining estimated GFR and incident end-stage renal disease. To date, none of the major trials with inhibitors of dipeptidyl peptidase-4 (DPP-4) or agonists of glucagon-like peptide-1 (GLP-1) receptor [128,129] have been proposed. In addition, endocannabinoids have also been proposed as GLUT2 inhibitors [130].

### 5.2. Renal Cancers

Some studies have suggested that, in several cancers, anticancer therapy may act through the inhibition of glucose transporters [131]. Unfortunately, little is known on this mode of therapy in renal cancers.

## 6. Summary

The kidneys play an important role in glucose homeostasis. They reabsorb glucose from urine and may increase its level in the circulating plasma by gluconeogenesis. The renal processes involve a range of GLUT and SGLT glucose transporters. Disturbances in their expression and/or functions may cause the onset of renal diseases and vice versa. Antidiabetic drugs are known to inhibit the activity of glucose transporters, especially SGLT2, and this may be an effective therapy against renal diseases, such as diabetic nephropathy, chronic kidney disease, and end-stage renal disease.

## Data Availability

All data presented in article are available in cited articles.

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
