# Peer review of "Human Glucose Transporters in Renal Glucose Homeostasis"

_ijms, 2021, doi:10.3390/ijms222413522_

Round 1

Reviewer 1 Report

Line 84 and 86 in page 2: MFS or MPS?

The reviewer do not understand the rationale between changes in glucose transporters in ARF and GFR decline in the section 4.5.

Due to the weight in the literature, the reviewer suggest to put a word “cancer” in the Abstract.

The flow through the line 533 to 537 seems to be discrepant; lowering SNGFR and slowing progression of CKD.

Author Response

Dear Reviewer

Reviewer 2 Report

In this work, Sedzikowska et al. perform an update on human renal glucose transporters. This interesting review is comprehensive and the two tables sum up the present knowledge on GLUT and SGLT expression in renal tubules. This work reports also phenotype and glucose transporter expression in different diseases including diabetes and metabolic syndrome and also  in patients with glucose transporter inactivating mutations. Last, the effect of SGLT2 inhibitors provides a significant additional value to this review. 

To my view, Section 5 "role of glucose transporters as potential targets in kidney disease" should be rather entitled "effect of SGLT2 inhibitors ...." in order to strengthen this new field and as incretin based drugs are out of the scope of this review (data on cancer are not very useful) . Furthermore, the data on SGLT2 inhibitors should be increased if possible noteworthy the result of trials on glycemia control and glycosuria magnitude and if available data on tubular glucose transporters and metabolism (neoglucogenesis) under treatment. 

Line 51: hydroxylation of vitamin D3

Line 54 amounts of glucose

Line 56: 0.3 - 16 ml/min of urine is produced

Line 90: SCN means?

Line 551: no verb in the sentence

Line 561 Renal pathological processes

Author Response

Dear Reviewer
